# Addison’s Disease in the Course of Recurrent Microangiopathic Antiphospholipid Syndrome—A Clinical Presentation and Review of the Literature

**DOI:** 10.3390/medicina59010004

**Published:** 2022-12-20

**Authors:** Małgorzata Grabarczyk, Marta Gorczyca, Paweł Cieślik, Antoni Hrycek, Michał Holecki

**Affiliations:** 1Student Scientific Society at the Department of Internal Medicine, Autoimmune and Metabolic Diseases, Faculty of Medical Sciences in Katowice, Medical University of Silesia, 40-055 Katowice, Poland; 2Department of Internal Medicine, Autoimmune and Metabolic Diseases, Faculty of Medical Sciences in Katowice, Medical University of Silesia, 40-055 Katowice, Poland

**Keywords:** adrenal insufficiency, microangiopathic antiphospholipid syndrome, clinical presentation

## Abstract

The article presents a male patient with adrenocortical insufficiency in the course of antiphospholipid syndrome (APS). It also describes recurrent exacerbations of his clinical status, characteristic of microangiopathic antiphospholipid syndrome (MAPS) which had been misdiagnosed as a disseminated intravascular coagulopathy (DIC) syndrome due to sepsis with multi-organ failure, including heart, kidneys, and liver. Issues related to pathogenesis, clinical symptoms, differential diagnosis, and treatment of APS in the context of presently distinguished subtypes of this syndrome have been addressed. The role of vascular endothelial cell activation and the influence of coagulation patterns on the development of APS continuum clinical symptoms have also been mentioned. In addition, this paper highlights that the diagnosis of APS should be considered in patients with adrenal insufficiency and abdominal pain, even without any prior history of thromboembolic diseases, as well as in the course of DIC, especially without predisposing factors.

## 1. Introduction

Antiphospholipid syndrome (APS) is an autoimmune thrombophilia, associated with the presence of antiphospholipid antibodies (aPL) against plasma phospholipid binding proteins, mainly to beta2-glycoprotein I (β2GPI) [1,2,3]. APS can occur as a primary syndrome without any other underlying disease, or it can be associated with systemic lupus erythematosus (SLE) or, more rarely, with some other autoimmune diseases [2,3]. Diagnosis of APS, according to the 2006 criteria, is based on the simultaneous presence of at least one clinical symptom and one laboratory finding [1,2,3]. Clinical symptoms include vascular thrombosis and pregnancy morbidity [1,2,3]. Laboratory criteria include the presence of aPL antibodies, such as lupus anticoagulant (LA), anticardiolipin antibodies (aCL) and antibodies to beta2-glycoprotein I (β2GPI), both IgG and IgM isotypes [1,2,3].

The aPL antibodies identify phospholipids only after their binding to β2GPI [4,5]. The aPL/β2GPI complexes then induce the development of adhesive, proinflammatory and prothrombotic effects on vascular endothelium cells, via multi-dimensional mechanisms [4,5]. Although in vitro studies elucidate some of the pathogenetic mechanisms of the aPL antibodies, it remains unanswered why only some patients suffer from thrombotic complications [6,7]. In addition, many patients with the aPL antibodies have clinical symptoms indicative of APS; however, they do not meet the classification criteria for this syndrome [7,8,9]. Thus, some emerging concepts treat APS not as single clinical syndrome but rather as a group of various aPL antibody-associated disorders [8]. The proposed APS classification includes a probable APS (pre-APS), a seronegative APS (SNAPS), a microangiopathic APS (MAPS) and a catastrophic APS (CAPS) [6,7]. Pre-APS can be diagnosed in patients with occlusion of small blood vessels of at least one organ or tissue and the presence of disorders, which can precede the APS development, such as livedo reticularis, chorea, thrombocytopenia, spontaneous abortion or valvular heart disease [6,7,10]. SNAPS includes patients with arterial or venous thrombosis, initially without any detectable aPL antibodies, but whose presence can be detected several months after the incident [7,10]. A diagnosis of MAPS can be made in patients with thrombotic microangiopathy (TMA), or thrombotic microangiopathic hemolytic anemia (TMHA) when accompanied by non-immune, hemolytic anemia with fragmented red blood cells (schistocytes) [8,9] and aPL antibodies [8,9,10,11]. TMA is found in the microvessels of the glomeruli, skin, retina, intestines, liver or lungs [11]. TMHA may also be seen in patients with SLE, Thrombotic Thrombocytopenic Purpura (TTP), Hemolytic–Uremic Syndrome (HUS), HELLP syndrome, malignant hypertension, postpartum renal failure or preeclampsia [11]. The presence of TMA represents the main difference between MAPS and APS, in which thrombosis occurs in large blood vessels and aPL can be generated by endothelial damage [11]. CAPS is characterized by a systemic TMA of small vessels, associated with cytokine release from damaged endothelial cells [4,12]. Clinical symptoms of CAPS develop very rapidly, due to the accompanying multi-organ failure, usually involving the kidneys, lungs, brain and heart. CAPS is often associated with systemic inflammatory response syndrome (SIRS), disseminated intravascular coagulation (DIC), severe thrombocytopenia and hemolytic anemia [4,5,12].

Primary hypoadrenocorticism (Addison’s disease; AI) is the result of destruction of the adrenal cortex and is manifested by the deficiency of gluco- and mineralocorticoids [13,14]. Most of cases are caused by an autoimmune process. The other causes include infectious diseases, usually caused by Mycobacterium tuberculosis, Meningococcus (Waterhouse–Friderichsen syndrome), Pseudomonas aeruginosa and unusual disorders such as APS [13,14,15]. Espinosa et al. revealed a causal relationship between aPL antibodies and the development of AI, which is the most common endocrine syndrome in APS [16,17]. AI was reported in 0.4% of patients with APS [17], and in more than 35% of patients with AI it is a first clinical symptom in the course of APS [16,17]. aPL antibodies are considered the main risk factor leading to adrenal hemorrhage [18,19]. Histopathological examination of adrenal glands revealed vasculitis in all APS patients with AI, while bilateral adrenal hemorrhage with adrenal vein thrombosis is found in about 59% of patients [15,16,17]. Exceptional blood supply to the adrenal glands, with increased vascularity supplied by three arteries, and limited venous drainage with only one vein, seems to be a predisposing factor for the development of adrenal vessel thrombosis [15,16,17]. Another proposed mechanism may be related to direct damage of zona fasciculata cells containing endosomes with lysobisphosphatidic acid, which is a target of aPL antibodies [15,17]. Lysosomal proteases released in reaction with the aPL antibodies activate the surrounding endothelial cells, leading to the development of thrombosis [15]. So far, there is no clear explanation for adrenal hemorrhages in the course of APS, which primarily predisposes one to thrombosis [17,19].

The aim of the study was to present diagnostic difficulties related to the coexistence of two rare diseases, which are APS and AI. We present a patient with various multi-organ symptoms, indicating that APS is not a one single disease but a complex disorder of an APS continuum. In addition, we address the diagnostic difficulties of APS classification faced by clinicians. We discuss the difficulties in interpreting clinical symptoms and indicate that the diagnosis of APS should be considered in AI patients with abdominal pain, even without previous thromboembolic episodes, and in patients with DIC, especially without any predisposing factors.

## 2. Case Report

A 54-year-old man was admitted to the hospital in October 2013, with persistent symptoms of general fatigue, loss of appetite, nausea, flatulence, no defecation disorders, followed by intermittent abdominal pain, and elevated body temperature to 38.2 °C. The medical history concerns an episode of venous thrombosis of the lower limbs 20 years ago, and primary AI diagnosed in 2006 in the course of Waterhouse–Friderichsen syndrome, due to probable meningococcal septic shock and DIC. Treatment with hydrocortisone at a single dose of 10 mg/d and fludrocortisone 0.05 mg/d was started. Unfortunately, the patient did not increase the dose of steroids, despite the endocrinologist’s recommendations from many years ago.

The physical examination revealed hyperpigmentation of the skin and oral mucosa, the abdomen was slightly tender on palpation and bowel movements were normal. In addition, a venous leg ulcer was present in the course of post-thrombotic syndrome. Laboratory tests were performed (Table 1), and the following results were found: normocytic anemia (Hb 80.00 g/L, reference range: 135–165 g/L) with schistocytes (0.8%) (Figure 1), thrombocytopenia (79 × 10^9^/L, reference range: 130–400 × 10^9^/L) and leukocytosis (11.18 × 10^9^/L, reference range: 4.00–10 × 10^9^/L). The direct Coombs’ test result was negative, whereas lactate dehydrogenase (LDH) activity was increased to 299 IU/L (reference range: <255 IU/L). Abnormal findings of the coagulation panel included a lack of thrombosis in the activated partial thromboplastin time test (APTT) and prolonged prothrombin time (PT; 24.5 s, reference range: 9.4–12.5 s). In addition, a positive result of the indirect serum antiglobulin test and increased concentration of D-dimer (4728 μg/L, reference range: <500 μg/L) were found. Markers of inflammation were elevated, including: ESR (146 mm/hr, reference range: 3–8 mm/h), fibrinogen concentrations (7.04 g/L, reference range: 2.00–3.93 g/L) and CRP level (219.45 mg/L, reference range: <5 mg/L). Liver function tests were moderately elevated. In addition, creatinine was increased (209.69 µmol/L, reference range: 64.05–95.31 μmol/L). Urinalysis (Table 1) revealed hematuria with 250 erythrocytes per μL and leukocyturia with 100 leukocytes per μL. The immunologic tests confirmed the presence of antinuclear antibodies (a titer of 1:320; homogenous-macular fluorescence type), and an antinuclear antibody (ANA) profile revealed presence of anti-centromere B protein antibodies (+++) and a lack of nDNA antibodies, as well as lack of autoantibodies against the adrenal gland. The anticardiolipin antibodies (aCL) of the IgG isotype had a high titer (47.4 GPLU/mL, reference range: <12.0 GPLU/mL), whereas aCL-IgM were within a normal range (1.6 MPLU/mL, reference range: <12.0 MPLU/mL); LA was present (LA-Screen: PTT 64 s, reference range: 26–36 s; PTT-LA 130 s, reference range: 33–42 s; DRVVT (Dilute Russell viper venom test): prolongated; LA-Confirm: positive; mixing study: no normalization) (Table 1). Moreover, protein C was 96% (reference range: 70–130%), and protein S was 69% (reference range: 60–140%).

Bacteriological and fungal cultures of urine and blood were negative. Microbiological profile of leg ulcer showed only slight (+) Staphylococcus aureus and sparse Klebsiella pneumonia (sensitive to most antibiotics) colonization.

Chest X-ray showed no abnormalities. Ultrasound and computed tomography of the abdominal cavity showed a dilated venous vessel in the peritoneal cavity (without signs of portal thrombosis in Color Doppler), retroperitoneal lymphadenopathy (up to 12 mm in diameter) and a normal biliary tract; the liver, spleen, pancreas, kidneys and adrenal glands were of the correct shape and size under homogeneous contrast enhancement.

In-hospital therapy, including intravenous (i.v.) hydrocortisone 200 mg/d, subcutaneous (s.c.) dalteparin 5000 IU/d and i.v. ciprofloxacin 400 mg/d resulted in immediate relief of abdominal pain and fatigue. Unfortunately, during the next 3 days of hospitalization, platelet count decreased to 20 × 10^9^/L (reference range: 130–400 × 10^9^/L).

Therefore, the treatment was modified, replacing hydrocortisone with methylprednisolone at a dose of 32 mg/d per os (p.o.), and dalteparin was discontinued due to the probable risk of heparin-induced thrombocytopenia (anti-heparin antibodies were not detected). At this point, all previously abnormal laboratory findings gradually improved. The patient was discharged home after 2 weeks of hospitalization. He was diagnosed with APS, AI in the course of the disease and post-thrombotic syndrome of the lower limbs.

During the outpatient follow-up visit, the patient was in good health, with no complaints or abnormalities in the physical examination. Control laboratory tests showed a decreased platelet count (78 × 10^9^/L, reference range: 130–400 × 10^9^/L) and prolonged APTT (77.7 s, reference range: 25.4–36.9 s). Presence of aCL was confirmed 12 weeks after initial diagnosis (aCL-IgM 58.4 GPLU/mL, reference range: <12.0 GPLU/mL; aCL-IgM <1.0 MPLU/mL, reference range: <12.0 MPLU/mL). The presence of venous thrombosis and one of the five (LA, and aCL or β2GPI both IgG and IgM) types of aPL antibodies allowed for the diagnosis of APS [1], and a daily oral anticoagulant therapy was administered.

It should be noted that the patient was also hospitalized in another department, in August 2013, due to similar symptoms, including abdominal pain with vomiting and diarrhea, and low-grade fever. Subsequently, the cortisol level was 2.9 ng/mL while being treated with hydrocortisone at a single daily dose of 10 mg. In addition, laboratory tests (Table 1) showed anemia with thrombocytopenia and prolonged APTT and PT. Inflammatory markers, as well as liver and kidney function tests, were elevated. Bacterial and viral infections were excluded. During this hospitalization, the patient experienced an episode of tachyarrhythmia with concomitant hypotension. Echocardiography revealed lower wall dyskinesia, akinesia of the posterior and hypokinesia of the lateral wall of the left ventricle, as well as a reduced ejection fraction up to 38%. The following diagnoses were reported on discharge: DIC with AI, probable post-inflammatory cardiomyopathy and chronic kidney disease.

## 3. Discussion

The presented case illustrates an example of APS that was diagnosed after a 20-year course of the disease. The diagnosis of APS is an important step in clinical management, as it allows the explanation of the etiopathogenesis of all clinical symptoms that occurred in this case. In our opinion, AI should be related with APS, but not with septic shock or Waterhouse–Friderichsen syndrome. In addition, the deterioration of the patient’s clinical condition in August 2013 could be related to the exacerbation of AI. At that time, the patient reported the typical clinical symptoms of AI, such as low-grade fever, general weakness, hypotension, lack of appetite, nausea, vomiting and abdominal pain. It should be noted that AI is most common endocrine disorder in APS and should be suspected in all APS patients who complain of abdominal pain [15,18,19].

In the presented case, we analyzed the recurrent exacerbations of the patient’s clinical condition, which was diagnosed as DIC, in the course of sepsis associated with hepatic, renal, and heart failure, in 2006, and in August 2013. Consequently, such a development of the disorder went beyond the diagnosis of APS, for which diagnostic criteria include only medium or large vessel thrombosis [1]. In particular, the progression of multi-organ failure can be explained by small vessel thrombosis, even without histopathological confirmation of the TMA. TMA is characterized by 2 subtypes of APS: CAPS and MAPS, which should be considered in the differential diagnosis [6]. The presented case, however, indicates the diagnostic challenge of MAPS and CAPS, especially related to the diagnosis of TMA based on histopathological examination, because the decision to perform a biopsy in patients with severe clinical conditions resembling sepsis or septic shock with severe thrombocytopenia and suspected DIC is very difficult.

MAPS is characterized by TMA-related organ damage [20,21]. Similarly, in the presented case, TMA was associated with renal, hepatic and heart failure. The development of heart failure depends on the number of small vessels closed by fibrin deposits, and on the size of the myocardium, with decreased perfusion [22]. Indeed, the results of echocardiography of the presented patient confirmed hypokinesis of the left ventricle and reduced ejection fraction. However, no valvular lesions, common in APS [23], were found, despite a 20-year course of his disease. We also believe that transient renal failure may be associated with TMA and the development of vasculitis, which rarely leads to chronic renal disease [22]. Moreover, liver failure, which is the most common manifestation of abdominal MAPS, was probably also the result of TMA [12].

CAPS, on the other hand, is characterized by the rapid development of clinical symptoms imitating sepsis, which can lead to death in 33,3% of patients within a few days [4,5,7,24]. These symptoms relate to renal, respiratory and cardiac failure, with severe thrombocytopenia, hemolytic anemia and SIRS or DIC [4,5,7,12,24]. In our opinion, the diagnosis of CAPS cannot be confirmed solely on the clinical picture [1]. In particular, the development of the complex somatic disorder in the presented case did not resemble sepsis. There were also no signs of ARDS or SIRS, and the absence of protein C and protein S and antithrombin deficiencies excluded the diagnosis of DIC. On the other hand, screening for aPL is necessary for all patients with DIC features when they do not have risk factors such as surgery, infections or trauma [16,25].

Other causes of TMA development should be mentioned, including HUS, escherichia coli shiga toxins (STEC), genetic complement abnormalities (atypical HUS) and secondary to drug-induced or with TTP D-HUS [11]. However, in the middle-aged man, clinical recurrent exacerbations, without bloody diarrhea, psychiatric disturbances or progressive kidney disease, are likely to exclude the presence of these diseases. However, the lack of determination of CH50, C3, C4 or ADAMTS13 is a significant limitation of our study.

Particular attention should be paid to the exacerbations of the patient’s clinical condition, requiring several hospitalizations accompanied by mild fever, nausea, vomiting, weakness and hypotension. The concentration of CRP and the ESR rate were increased, as well as a prolonged PT and APTT, without in vitro clotting, but with a decrease in the number of platelets and an increase in D-dimer serum level, dysfunction of the heart, liver and kidneys, secondary to in vivo clotting. The observed increase in both serum transaminases and creatinine as well as microscopic hematuria were only transient and, importantly, always without any noticeable infection. Clinical improvement occurred after increasing the dose of substitution steroids, and was too fast for commonly used antibiotics, e.g., ciprofloxacin. Therefore, according to our assessment, the cause of exacerbations was not infection but MAPS with thrombi in the microcapillaries of the kidneys and other organs. MAPS was also the cause of increased demand for steroid hormones; so the observed abdominal pain, hypotonia and mild fever were an expression of AI exacerbation [18].

## 4. Conclusions

In conclusion, it should be stated that this patient presented with AI in the course of APS and with recurrent MAPS, and this illustrates a clinical scenario covering a wide spectrum of disorders typical for APS. The presented case indicates the need to suspect APS syndrome in all patients with AI, especially those with abdominal pain [16,17], and in all patients with suspected DIC, especially without predisposing factors [25].

## Figures and Tables

**Figure 1 medicina-59-00004-f001:**
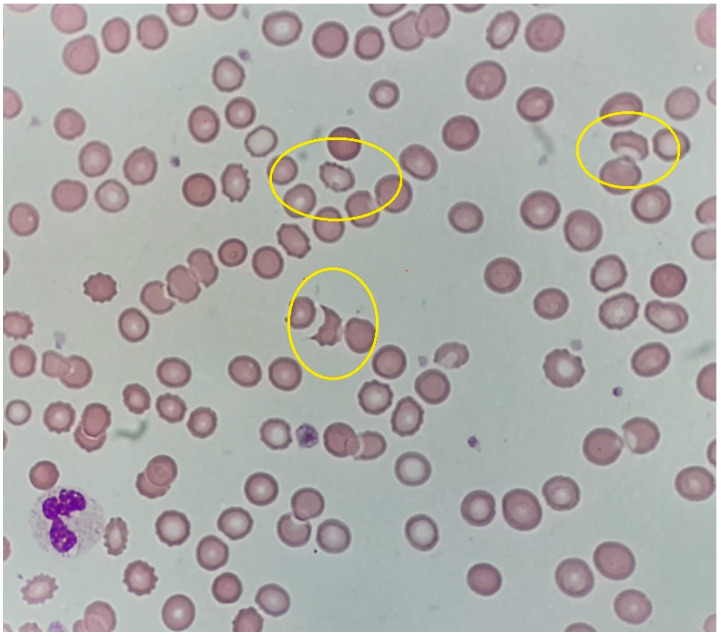
Schistocytes in peripheral blood smear (H&E, 1000×).

**Table 1 medicina-59-00004-t001:** Laboratory test results—blood morphology, urinalysis, biochemical and immunological test results.

Date	27 August 2013	16 September 2013	23 October 2013	29 October 2013	7 November 2013	14 January 2014	Range
Blood count							
Hb (g/L)	133	90	80	82	87	114	135–165
Ht (%)	40	26.6	25.7	25.3	27.4	36.8	40–53
RBC (1012/L)	4.57	3.12	2.89	2.83	3.07	3.78	4.2–5.7
WBC (109/L)	6.7	5.96	11.18	13.23	13.64	6.51	4.0–10.0
PLT (109/L)	51	115	79	20	82	90	130–400
Coagulation test							
aPTT (s)	lack	118.3	lack	108.5	207.6	67.2	25.4–36.9
INR	2.34	1.56	2.17	1.78	1.75	1.2	0.8–1.2
PT (%)		59	33	45	46	78	80–120
Fibrinogen (g/L)		7.87	7.04	2.92	4.68	2.21	2.0–3.93
Antithrombin III (%)			62	54	88		75–120
D-dimers (μg/L)		5210	4728	16,250	1084	1697	<500
Urine microscopic exam							
pH	5.0	-	5.0	5.0	7.0	5.0	4.8–7.4
specific gravity (g/dm^3^)	1.025	-	1.015	1.015	1.005	1.020	1.016–1.022
glucose (mg/dL)	neg.	-	neg.	neg.	neg.	neg.	<30
protein (mg/dL)	neg.	-	75.0	25.0	neg.	25.0	<10
leucocytes (cells/μL)	-	-	100	neg.	neg.	neg.	<10
erythrocytes (cells/μL)	-	-	250	50	150	10	<5
Urine sediment							
WBCs (cells/HPF)	3–5	-	20–25	4–6	0–1	0–1	0–5
Normal RBCs (cells/HPF)	1–2	-	0–1	0–1	2–5	0–1	0–2
Dysmorphic RBCs (cells/HPF)	0–1	-	massive	5–8	0	0–1	
Serum chemistry							
CRP (mg/L)	348.8	20.5	219.5	42.3	50.0	4.3	<5
Glucose (mmol/L)	6.28	8.61	5.72	6.17	5.94	5.06	3.9–5.5
Na (mmol/L)	127.8	131.4	136	140	135	142	136–156
K (mmol/L)	4.9	4.1	2.9	3.6	3.7	3.6	3.5–5.1
AST (U/L)	53.4	129	56	110	28	10	<35
ALT (U/L)	22.7	60	23	66	35	11	<45
GGTP (U/mL)	34	123	22	122	147	17	<55
ALP (U/L)	125	211	105			36	38–126
Bilirubin (μmol/L)	13.2	12.8	14.9	8.9	13.2	10.8	3.4–17.0
Creatinine (μmol/L)	297.9	170.6	209.7	159.1	87.5	102.5	64.1–95.3
LDH (U/L)		360	299				<248
ACTH (pg/mL)						11.6	<46
Cortisol (μg/dL)	2.9		0.25			2.93	4.3–22.4
Immunological data							
ACL-IgM (MPLU/mL)			1.6			<1.0	<12
ACL-IgG (GPLU/mL)			47.4			58.4	<12
LA			presence				
ANA index			positive				
ANA titer			1:320				
NA pattern			homogenous-macular				
centromere B protein			+++				
RNP/Sm, Sm, SSA, Ro-52, SSB, Scl-70, PM-Scl, Jo-1, PCNA, dsDNA, nucleosome, histons, ribosomal-P-protein, AMA-M2			absent				
nDNA (U/mL)			90.7				<100

Hb—Hemoglobin, Ht—Hematocrit, RBC—red blood cells, WBC—white blood cells, PLT—Platelets, aPTT—Activated Partial Thromboplastin Time, lack—lack of thrombosis, INR/PT—International Normalized Ratio/Prothrombin time, neg.—negative, cells/HPF—cells per high power field, CRP—C-reactive protein, Na—sodium, K—potassium, AST—aspartate transaminase, ALT—alanine transaminase, GGTP—Gamma-glutamyltransferase, ALP—alkaline phosphatase, LDH—lactate dehydrogenase, ACTH—adrenocorticotropic hormone, ACL—anticardiolipin antibodies, LA—Lupus anticoagulant, ANA—antinuclear antibodies, MPLU—IgM phospholipid unit, GPLU—IgG phospholipid unit, NA pattern—nuclear antigens fluorescence pattern, RNP/Sm—ribonucleoprotein/Smith’s protein, SSA—Sjögren’s syndrome-related antigen A, SSB—Sjögren’s syndrome-related antigen B, PCNA—proliferating cell nuclear antigen, +++—very high antibody titer.

## Data Availability

Not applicable.

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
