# Peer review of "Addison’s Disease in the Course of Recurrent Microangiopathic Antiphospholipid Syndrome—A Clinical Presentation and Review of the Literature"

_medicina, 2022, doi:10.3390/medicina59010004_

Round 1

Reviewer 1 Report (Previous Reviewer 2)

The manuscript has been revised well. I think this manuscript will be acceptable.

Author Response

Thank you once again for your time and insightful comments, which enrich the article and provide the Readers with a more complete picture of the presented clinical case.

Reviewer 2 Report (New Reviewer)

Please see the attached review.

Author Response

Thank you very much for your time and insightful comments, which significantly enrich the article and provide Readers with a more complete picture of the presented clinical case.

The answers to the comments:

  1. Introduction: lack of the study aim.
    The aim of the work has been included in the introduction.
  2. “Moreover, AI can also be the first clinical symptom of APS, in above 35% of patients”- unclear.
    We have made the appropriate correction.
  3. The case presentation is not fully understandable.
    a. “His past medical history is significant for an episode of lower extremities, venous thrombosis 20 years ago and primary AI, diagnosed in 2006, caused by Waterhouse-Friderchsen’s syndrome, in process of septic shock with associated DIC syndrome.”- unclear description of the episode; was the hydrocortisone treatment implemented?
    We have made the appropriate correction and we have provided information about the treatment.
    b. “It should be noted that the patient had also been hospitalized in another Department, in August 2013, due to similar symptoms, including abdominal pain, associated with vomiting and diarrhea, and low-grade fever. The following diagnoses were reported on discharge: DIC with AI, cardiomyopathy probably post-inflammatory, and chronic kidney disease.”- what were the cortisol levels?; again, please add an information about the hydrocortisone treatment.
    We have provided treatment and cortisol levels information.
  4. “Urinalysis revealed abundant microscopic hematuria and few leukocytes.”- the results should be specific.
    We have included the specific results in Table 1, and in the main text.
  5. Table 1: what were the blood glucose levels?
    We have included glucose levels, thank you for your attention.
  6. Conclusions: “Therefore, the authors propose the possibility of including some new diagnostic markers of vascular endothelial activation, such as pentraxin 3, to expedite diagnostic workup [25]. However, this potential noninvasive strategy needs to be examined in future prospective clinical studies.”- this statement does not have any background in the case presentation and was not mentioned in the Discussion.
    Indeed, the information about pentraxin 3 is not relevant here and has been removed.
  7. References: limited references regarding AI.
    The references regarding AI have been corrected.

Round 2

Reviewer 2 Report (New Reviewer)

Please see the attached review.

Author Response

Thank you very much for your time and insightful comments, which significantly enrich the article and provide Readers with a more complete picture of the presented clinical case.

The answers to the comments:

  1. “Treatment with hydrocortisone at a daily dose of 0.01 and fludrocortisone 0.05 mg/d was initiated.” There is a probable misspelling- it should be 0.01 g/d or rather 10 mg/d. Was it given in one or two doses?
    The dose of hydrocortisone was change in the manuscript.

  2. “Then, the cortisol level was 2.9 ng/mL despite treatment with hydrocortisone at a daily dose of 10 mg.” Actually, if the patient has been treated with hydrocortisone the cortisol level (probably morning) will be low, so it should be rather e.g. while treated with hydrocortisone.
    We have made the appropriate correction.

  3. There is lack of information if the patient had been informed about the necessity of increasing the doses of hydrocortisone in a case of an acute illness, and if the patient actually had increased the doses.
    We have included the relevant information in the manuscript.

  4. I cannot see the date of the presented hospitalisation. It would be good to add this information to have a time comparison with the past hospital admissions (2006, 2013)
    We have added the date of presented hospitalisation in the manuscript.

  5. I would recommend adding the present diagnosis at the end of the case presentation to make it clear for the readers, e.g. At present, the patient’s diagnosis is…
    We added the diagnosis of present hospitalization in the manuscript.

  6. “Moreover, the presented case points out the diagnostic challenge need for modification of diagnostic criteria of MAPS and CAPS, particularly the ones related to diagnosing TMA, based on histopathology examination. A decision to perform a biopsy in patients with severe clinical conditions, resembling sepsis or septic shock, with serious thrombocytopenia, and suspected DIC is very difficult.” In my opinion this part of the conclusions belongs rather to a Discussion.
    We moved the sentences to the Discussion.

Moreover, we improved the english language of the manuscript.

Thank you once again for your time and insightful comments. We agree with all the comments and we are glad that the manuscript is becoming more and more transparent for Readers.

This manuscript is a resubmission of an earlier submission. The following is a list of the peer review reports and author responses from that submission.

Round 1

Reviewer 1 Report

I read with interest your article. Unfortunately, I did not find interest. The references are not as relevant as it must be, for example you even did not cite the international register. I also find that it is not so clear for authors what really happened and the link between every involvement and for example you still believe that a renal lupus vasculitis occurred. The importance of low arterial blood pressure in the diagnosis of adrenal insufficiency for example. 

Reviewer 2 Report

Comments to the Author
Thank you for asking me to review this
Addison’s disease due to microangiopathic APS written by Dr. Grabarczyk and colleagues.  It is interesting that this case indicates adrenocortical insufficiency might be an important symptom of APS. On the other hand, I think there are several important concerns in diagnosing as microangiopathic APS in this case.

Major comment:

1.Please report the values for the lupus anticoagulant, B2 glycoprotein, anti-prothrombin antibody  and haptoglobin clearly, along with the reference ranges. In addition, please add the titers of lupus anticoagulant, B2 glycoprotein and anti-cardiolipin IgG and IgM antibody of repeated tests after 12 weeks.

2. Please report percentage of schistocytes in the blood smear.

3. Please note the titer of complements (CH50, C3 and C4). If possible, please make tables including the laboratory data. Based on a previous study, hypocomplementemia was found in 47% of primary APS.( Ref. Ramos-Casals M, Campoamor MT, Chamorro A, et al. Hypocomplementemia in systemic lupus erythematosus and primary antiphospholipid syndrome: prevalence and clinical significance in 667 patients. Lupus 2004; 13: 777–783.)

4. The author mentioned this case was suggested as heparin-induced thrombocytopenia. Please note the titer of anti-HIT antibody with reference ranges.
5. Please mention the process of differentiating other conditions which cause thrombotic microangiopathy, such as STEC (shiga toxin-prodicing Escherichia coli) hemolytic uremic syndrome (HUS), atypical HUS and secondary (autoimmune or drug-induced) TMA. Please report the values of ADAMTS 13 activity, ADAMTS 13 inhibitor and blood culture including shiga toxin.

6. I’d like to know why CRP is markedly elevated in this case. Please explain the cause and rule out infectious conditions.

7. Please show the images of CT scan, brain MRI or ultrasound of cardia and lower extrimities to rule out large arterial or venous thrombosis.

8. Please include the case report part applying the classification criteria of anti-phospholipid syndrome in this case. (Ref. Miyakis S, Lockshin MD, Atsumi T, Branch DW, Brey RL, Cervera R, Derksen RH, DE Groot PG, Koike T, Meroni PL, Reber G, Shoenfeld Y, Tincani A, Vlachoyiannopoulos PG, Krilis SA. International consensus statement on an update of the classification criteria for definite antiphospholipid syndrome (APS). J Thromb Haemost. 2006 Feb;4(2):295-306.)

9.His past medical story indicated venous thrombosis. Thrombotic conditions such as protein S or C deficiency should be ruled out.

10. The author wrote that a diagnosis of TMA was made in this case. Have you consider choosing the treatment option of plasma exchange in this case?  

11. This case was not performed histopathological examination exhibiting thrombotic microangiopathy or endothelial cell injury. Was it difficult in this case?